# A Mutation Threshold for Cooperative Takeover

**DOI:** 10.3390/life12020254

**Published:** 2022-02-08

**Authors:** Alexandre Champagne-Ruel, Paul Charbonneau

**Affiliations:** Département de Physique, Université de Montréal, QC H2V 0B3, 2900 Edouard Montpetit Blvd, Montreal, QC H3T 1J4, Canada; paul.charbonneau@umontreal.ca

**Keywords:** origin of life, prisoner’s dilemma, cooperation, stochasticity, critical phenomena, thresholds

## Abstract

One of the leading theories for the origin of life includes the hypothesis according to which life would have evolved as cooperative networks of molecules. Explaining cooperation—and particularly, its emergence in favoring the evolution of life-bearing molecules—is thus a key element in describing the transition from nonlife to life. Using agent-based modeling of the iterated prisoner’s dilemma, we investigate the emergence of cooperative behavior in a stochastic and spatially extended setting and characterize the effects of inheritance and variability. We demonstrate that there is a mutation threshold above which cooperation is—counterintuitively—selected, which drives a dramatic and robust cooperative takeover of the whole system sustained consistently up to the error catastrophe, in a manner reminiscent of typical phase transition phenomena in statistical physics. Moreover, our results also imply that one of the simplest conditional cooperative strategies, “Tit-for-Tat”, plays a key role in the emergence of cooperative behavior required for the origin of life.

## 1. Introduction

Attempts to explain the sequence of events leading to the origin of life have elicited much research over the years [1,2,3,4,5,6,7,8,9,10]. One of the leading theories includes the hypothesis according to which life would have evolved as cooperative networks of molecules [11,12,13]. In the context of this theory, two main elements however remain to be adequately described: a concrete specification of the physicochemical reality of such networks, and the means by which those would have evolved. Specifically, if we consider that life arose through the cooperation of life-bearing molecules, then explaining the dynamics that led to the evolution of such networks is a priority [14].

While Darwinism explains how macro-complexity derives from micro-complexity, it does not explain how the latter appears in the first place [15]. It is thus thought that the RNA world may not have been the earliest genetic system, favoring instead a system of simpler units having the capacity to encode and generate information through selection without formal replication—where candidates include polypeptides, RNA-like polymers, and lipids [14]. This begs the question of the origin of evolution: when does chemical kinetics become evolutionary dynamics [16]? In other words, a necessary *explanans* of origin of life theories lies in this chemical phase [15], a gray zone between non-life and simple life which required kinetic competition between information units prior to the advent of Darwinian evolution [17].

Many observations also suggest that Darwinian behavior is rooted in chemistry. Since it was observed in 1960 that the Qβ bacteriophage displayed Darwinian behavior, many other in vitro procedures have shown that other nucleic acid systems are also Darwinian [15]. Voytek and Joyce, for instance, noted that the biological phenomenon of competitive exclusion also applies at the chemical level to RNA enzymes [18], while Adamala and Szostak have investigated how changes in the composition of the membrane of protocell vesicles caused by a simple catalyst enable the origin of selection and competition between protocells [19]. More recently, it has been suggested that both prebiotic evolution and the evolution of biological systems follow similar equations, and Yeates et al. made this connection explicit [20]. It is thought that this “pre-life” phase would have taken the form of active monomers generative of information, and that the replication rate exceeding a threshold value would have elicited a transition sharing characteristics with phase transition phenomena [16].

Building on the concept of cooperating networks, Damer and Deamer has proposed that protocells undergo a primitive version of Darwinian selection for stability, termed combinatorial selection, and are solely driven by physical and chemical forces at the molecular level. Gel aggregates subjected to wet-dry cycles would enable protocell-to-protocell interaction, leading to a form of network selection [21,22], and would thus constitute the unit that Woese and Fox anticipated to be the ancestor of procaryotes and eucaryotes, termed the progenote [23]. Laboratory studies also support this view of a collectivist origin of biology presumed to give rise to autocatalyzed metabolic cycles: previous work on the interaction between RNA and lipid membranes reveal that oligomers bind stably to the liposome surface—a property that isn’t shared by individual RNA molecules [24,25]. Those results suggest that selection processes in the prebiotic world might yield RNA consortia, thereby accounting for the colocalization of proteins, RNA, and membrane as a single unit. These collective phenomena can all be viewed as instances of cooperative behavior at the chemical level.

Game theory provides the framework of choice to study how cooperation can emerge in competitive environments, and has inspired much theoretical and experimental work on this question. It has been used successfully in a wide array of problems [26,27,28], from ecology [29] to computational neuroscience [30], and is increasingly being applied in the context of prebiotic chemistry. If Eigen already considered prebiotic evolution to have properties of games, it has become recently even more evident that game theory is relevant for biophysics and biochemistry [31]. Evolution is almost always the co-evolution of the organisms and their environment [31] over dynamic fitness landscapes [32], which makes game theory is the appropriate framework for an analysis of such processes.

Examples of applications of game theory at the molecular level include multiple analyses where viruses are considered as players [33] and subsequent work on the Φ6 virus model system, HLSV, and TMV viruses, and on sets of phages [34]. Recent observations have also been made of RNA and autocatalytic sets that display competition, inheritance, and cooperative behavior [35,36,37]. It is thought that RNA polymers could thus meet the criteria for behavioral chemistry, make “choices” based on their environment and self-contained information [38] and more generally that macromolecules—RNA, DNA, and proteins altogether—could be regarded as players where strategies derive from their state and properties [34]. Partially unfolded molecules [38] and chemical networks [14] would possess a type of “memory” and respond to environmental conditions, subsequently adopting different conformations leading to differentiated behavior that would be subjected to natural selection, possibly through the use of “signals”—a feature distinguishing biochemistry from abiotic chemistry [39]. Formation of protein complexes have identically been studied as games [40] and genes considered as players [34]. Recent laboratory experiments have also contributed to establishing that game theory applies at the chemical level in systems where de novo synthesis takes place (i.e., reproduction instead of replication), and where kinetic selective forces apply [20].

It is thus becoming increasingly clear that a *chemical game theory* can shed some light on the dynamics of macromolecular interactions, and provide insight on the early emergence of cooperation, the latter being one key idea in understanding how individual entities can be spontaneously brought together into the formation of more complex structures where the interaction of the components with each other promotes the sustenance of the latter. While in biology cooperation is often explained away using references to concepts such as kin selection [41], this process cannot be correspondingly applied to cases where individuals possess no agency. Origin of life theories, insofar as they seek to provide a detailed *explanans* of the phenomenon, must adequately conceptualize how individual components subject to Darwinian principles can *cooperate* into forming autocatalytic sets [11,12,42], replicating another ribozyme without repayment [3,8,43], or forming whatever early proto-metabolism is required to jump-start the emergence of more complex lifeforms [44]. Cooperation is so interweaved into the fabric of complex life—from cell-cell communication and biofilm formation in microbes [45] to countless examples in the ecological and social sciences [46,47,48,49,50,51]—that it seems unthinkable to even consider putting forward a theory of the origin of life that wouldn’t account for the emergence of cooperative behavior in the first place. As Frick and Schuster eloquently put it: clearly, cells belonging to the same organism should not compete against one another [52].

There is thus a renewed interest in the importance of cooperation for the origin of life problem [17] and more generally, as Queller has suggested [53], in the context of major evolutionary transitions [54], which have involved increases in complexity and the establishment of new cooperative relationships [39]. There is notably ample evidence that cooperative molecular behavior, in general, constitutes an evolutionary advantage [15], thus being selected by evolution under the right circumstances. Specifically for the origin of life, cooperation stands as one of the three advantages of prebiotic networks put forward by Nghe et al., thus sparking the transition from chemistry to biology [14]. This view is also supported by laboratory experiments [55] and observations of RNA cooperative networks [14]. Explaining cooperation—and particularly, its emergence in the context of the origin of life—is clearly a required element in the description of the transition from non-life to life, and while much work has been done in explaining the origin of cooperation in high-order organisms, a convincing description of the way by which cooperative behavior emerges at the molecular level has yet to be elaborated.

## 2. The Prisoner’s Dilemma

The prisoner’s dilemma (PD) is precisely used in game theory to analyze the tension between rational, selfish behavior and altruistic cooperation [56,57,58] and has been put forth in the analysis of cooperative behavior in various natural and artificial systems. Two players each have the opportunity to either cooperate (C) or defect (D), without knowing in advance which strategy will be adopted by their opponent. Depending on the different combinations of moves chosen by each player, players will be rewarded with a score (Table 1). While mutual cooperation yields the maximum mean reward for the two players, exploitation (i.e., defecting when the opponent chooses to cooperate) leads to the maximum reward for the exploiting party and nil for the exploited player—mutual defection thereafter leading to the worst mean reward for both. This scoring scheme highlights the tension between selfish behavior that either leads to an immediate advantage or a poor outcome, on one hand, and cooperation as a compromise that leads to a reasonable outcome for both players on the other. Generalizing this principle leads to the formal constraint
(1)$$\begin{array}{c} T>R>P>S, \end{array}$$
with *R* being the reward, *T* the tentative payoff, *S* the sucker’s payoff, and *P* the punishment [59]. The PD as studied in game theory also applies to contexts where the players possess no capacity for rational reasoning, and is thus a useful framework in analyzing biochemical cooperation phenomena in the context of prebiotic chemistry.

A temporal extension of the PD known as the iterated prisoner’s dilemma (IPD) is the canonical model used to study the interaction between players. The game is played for several rounds, either for a definite number of times known to the players or by stopping the game with some finite probability. The players then follow a *strategy* that defines in advance their next move, taking—or not—into account the information available to them through the game at that time (e.g., their previous moves and those of the opponent). Basic strategies include the case where a player *always cooperates* (hereafter ALLC), *always defects* (ALLD), *shows mutual reciprocation* by mirroring the opponent’s previous move (TFT, for “Tit-for-Tat”), or *acts randomly* (RND).

Using agent-based modeling of the iterated prisoner’s dilemma, we investigate the emergence of cooperative behavior in a spatially extended setting and characterize the effects of inheritance and variability. We demonstrate that there is a mutation threshold above which cooperation is—counterintuitively—selected, which drives a dramatic and robust cooperative takeover of the whole system. Moreover, our results also imply that one of the simplest conditional cooperative strategies, “Tit-for-Tat”, plays a key role in the emergence of cooperative behavior required for the origin of life.

Rather than being considered a lattice-based simplified representation of a specific system, our simulations are best viewed in the context of constructive biology which seeks to identify universal patterns and mechanisms that are independent of specific physical/chemical contexts and substrates [10,60,61]. A cooperative takeover, by which we mean the rise to a predominance of reactive cooperating strategies in our simulations represents a prime example of emergent dynamics: namely, the appearance of novel features where cooperation is preeminent on global (“macroscopic”) scales that cannot be directly inferred or predicted, even from a complete knowledge of system behavior and interaction rules at small (“microscopic”) scales, with clear Darwinian dynamics that should, by all means, favor selfish behaviors. Furthermore, the emergence of cooperation as an emergent phenomenon witnessed in our simulations—and the wide range of the parameter space over which it occurs—suggests that the results obtained here are not strictly dependent on the specific implementation of the model.

## 3. Simulation Design and Methods

Spatial aspects of prebiotic settings are thought to have played a significant role in life’s early chemistry. Examples include montmorillonite clay and other mineral surfaces, which may have promoted the polymerization of activated mononucleotides [22]. Spatial games have likewise been the subject of much interest in the past decades (e.g., [62,63,64,65,66,67]). Our model implements IPD games on a 128×128 Cartesian lattice with periodic boundary conditions where each site is occupied by a player adopting one of the four strategies consisting of ALLC, ALLD, TFT, and RND. Strategies are first distributed on the lattice randomly in equal proportions (Figure 1A). At each new iteration of the model, every player plays IPD games of *M* rounds against every other player inside their Moore neighborhood, i.e., the eight sites surrounding them, and record their score (Figure 1B). In a second step, the strategy of every player of the lattice is replaced by the highest scoring strategy in their Moore neighborhood, if applicable (Figure 1C). This evaluation of the Moore neighborhood is randomized as not to introduce any spatial bias in the simulation: comparing scores between each strategy and its neighbors was done in a random order as not to favor a specific spatial pattern if two players obtained an even score. This process is then repeated for *T* iterations, which leads to fluctuations in strategy frequencies and the emergence of spatial self-organization dynamics. Simulations were carried out for T=500 iterations, which allowed a relaxation of the initial conditions sufficient for population dynamics specific to simulations parameters to materialize independently of the initial conditions. Finally, IPD game lengths of M=2000 moves were chosen to allow meaningful behavior in cases of very low error rates.

Figure 2 shows snapshots for the first fourteen iterations of the model for a simulation that includes ALLC, ALLD, TFT, and RND. In the vast majority of simulations carried out, including deterministic simulations initialized with the parameters specified above, the RND strategy is promptly evacuated from the lattice most of the time and was thus not further included in the initial strategy distribution. Deterministic simulations quickly (i.e., in a few tens iterations) reach a stationary state where TFT largely dominates. This is consistent with the known success of TFT [57,59].

The influence of stochasticity has also been discussed at length not only in relations to the origin of life problem and microbial life [68,69,70], but also in the context of game theory and the PD itself [67,71,72,73,74,75,76,77,78,79]. In the prebiotic context, it is generally agreed upon that an external supply of energy is required to drive non-equilibrium chemical reactions leading to the buildup of catalytic cycles and complex biomolecules [80,81,82,83,84,85,86]. However, strong external energy fluxes generally also imply strong thermodynamical fluctuations, which may be detrimental to the autonomous self-organization of complex prebiotic or early-life chemistry. Such fluctuations translate into either higher error rates in the interaction of early replicators, higher mutation rates, or both. For example, in an analysis of UV transmission in a prebiotic setting, the conclusions of Ranjan et al. are unambiguous: insofar as prebiotic chemistries are shown to be affected by UV irradiation, they must be demonstrated to either invoke a UV-shielded milieu or be so productive as to outpace UV degradation [87]. TFT’s success however notably relies on being placed in an environment where interactions are *reliable*—that is, where no “mistakes” are made by the players. Noisy environments, therefore, destroy any ongoing mutual cooperation between TFT strategies: as soon as a defection happens in the game, two TFT strategies playing against one another will enter a perpetual cycle of mutual defection.

If TFT’s success in error-free environments has now been known for some time, a central question relevant to the origin of life scenarios concerns the emergence of cooperative behavior in stochastic (i.e., noisy) environments. We have therefore implemented an error rate according to which the players will sometimes diverge from their strategy. For each IPD game, both players will thus play according to their assigned strategy, but will make mistakes (e.g., defect instead of cooperating with the other player) with a probability *p*. Varying this error rate impacts system behavior globally, both in terms of populations dynamics and in terms of the spatial structures produced. Error rates have been sampled from a lognormal distribution of mode $\hat{p}$, with logarithmic mean *m* and variance $s^{2}$ (Equation (Equation 2) in the Appendix A). Values of $p\in [10^{-6},5\times 10^{-1}]$ have been sampled, which represent values of the same order than presumed error rates in early replication mechanisms [88,89,90]. A detailed description of the algorithm used for evolving the systems and the implementation of stochastic elements is also included in the Appendix B.

In the context of the RNA world hypothesis, autocatalytic systems are thought to have evolved through the selection of both heritable and variable components. In a minimal model of early replicators, the error rate can thus be both subject to heritability and variability. We hence tested the impact on simulation dynamics of the heritability of error rates by including, for a subset of simulations, the heritability of the winning player’s error rate. We also introduced a mutation probability according to which a mutation, corresponding to a correction proportional to *s*, can occur when a player inherits another player’s error rate (c.f. Equation (Equation 4) of the Appendix B). Mutation rates $\mu \in \left[10^{-4},1\right]$ were sampled, which correspond to a few mutations per simulation up to some critical value typical of an *error catastrophe* [68]—a characteristic of replicating systems that has been compared to phase transition phenomena in statistical physics with replication error rate acting as the order parameter [91,92,93,94,95,96]. The presence of an error catastrophe is indeed predicted by the quasispecies theory: above the error threshold value, replication errors lead to populations being submerged by unfit individuals that decrease the overall fitness until the species goes extinct [68].

## 4. Results

### 4.1. Spatiality and Population Dynamics

Simulations evolved with strategies making mistakes (i.e., p>0) lead to a wide range of system behavior reflected in the spatial organization of strategies on the lattice. Two notable examples of such spatial configurations include regimes with time-varying fractal distributions of strategies (population evolution and final lattice state shown on Figure 3A,B) and clustering patterns (Figure 3C,D). While the evolution of the population reaches a dynamical equilibrium in the first case, the equilibrium becomes static in the second. In the latter case, the system’s dynamics leads to the formation of isolated domains of ALLC, which otherwise would not survive in exploitative environments, that can be sustained while surrounded by ALLD players. Likewise, ALLD strategies can persist in environments where TFT is present by preying on ALLC players. Spatiality is thus a required condition for such dynamics to emerge, and the score of surrounding parasites (e.g., ALLD players) cannot, in this case, exceed that of the central cooperator [59]. This has been observed when the duration of the interaction between players is long enough—i.e., game length M≳2000 PD moves—as to supersede the temporal dynamics of the simulation itself, hence this parameter value was conserved for all simulations.

In contrast, randomizing the players’ neighborhoods over the whole lattice—i.e., evolving environments that are “fully mixed”—prevents the formation of such organized spatial structures. Spatiality is thus an important determinant of population dynamics by allowing the survival of cooperative entities in adverse conditions by means of structure formation [97,98,99,100].

The final states of the lattice are in this sense completely independent of the (randomized) initial conditions. The trajectory of the system in phase space end up in the majority of cases on attractors, leading to spatial structures such as the ones shown in Figure 3D, or in limit cycles where emerges cyclical variations of the populations of each species (e.g., Figure 3A). The state of the lattice can thus either become fixed, as in the former case, or remain in dynamic equilibrium as in the latter.

### 4.2. Influence of the Error Rate

Many of these features associated with spatiality depend on the precise value of the strategies’ error rate: final population frequencies for ALLC, ALLD, and TFT relative to the most probable value of the initial error rate distribution $\hat{p}$ are shown on Figure 4A. An initial error rate was assigned to every player, drawn from a lognormal distribution (see Equation (Equation 2) in the Appendix B) with most probable value $\hat{p}$ shown in abscissa and shape parameter $s=5\times 10^{-1}$. Final population fractions are averaged over an ensemble of 20 simulations per sampled value of the error rate, and shaded areas indicate the standard deviation of the final population fractions. As $\hat{p}$ is varied from one simulation to the next, sharp transitions occur repeatedly in the population frequencies and their patterns of spatial distributions.

At very low error rates ($p∼10^{-6}$) cooperators have the upper hand: TFT quickly eliminates ALLD but can still become trapped in a mutual defection process with a nonzero probability when playing against itself, hence the final domination by ALLC. In contrast, at very high error rates (p∼0.5) TFT performs poorly against ALLD (i.e., it starts with cooperation, then follows ALLD’s moves—a rather poor behavior). Nothing then further prevents ALLD from eliminating ALLC and invading the whole lattice.

### 4.3. Heritability and Mutations of the Error Rate

Introducing heritability of error rates—i.e., the player losing the PD game adopting not only the opponent strategy but also its error rate—leads to increased variability in the outcome of simulations (Figure 4B). At very low error rates, TFT can now decrease its mean error rate through the successive selections of winning strategies; this allows an increase in TFT’s relative population frequency and a corresponding decrease in ALLC’s. Similarly, at very high error rates TFT’s decreased mean error rate allows it to be much more successful against ALLD. However, by also making error rates subject to *mutations* the system’s behavior again undergoes significant changes. Keeping all other parameters identical, a mutation probability $\mu =10^{-4}$ player${}^{-1}$
$t^{-1}$ is introduced (Figure 4C) which, when fulfilled, applies a multiplicative correction to a player’s error rate proportional to the shape parameter *s* of the initial error rates distribution (see Equation (Equation 4) in the Appendix B). While system behavior at $\hat{p}≲10^{-3}$ does not differ significantly from previous simulations, when $\hat{p}$ increases above this value there is a sharp transition in the relative frequency of TFT populations. This is explained by the fact that the extent to which TFT represents a successful strategy is inversely correlated with the error rate—that is, TFT lacks any error correction mechanism; if TFT plays against itself, the first error introduced in the game is bound to send both players into a spiral of mutual defection until the game ends [101]. Error rates are thus driven towards lower values when mutations are allowed, which ultimately leads to the complete invasion of the lattice by TFT.

This process was repeated while varying the mutation probability itself. For a fixed error rate initial distribution with $\hat{p}=10^{-4}$ and shape parameter $s=5\times 10^{-1}$, Figure 5A displays another statistical ensemble of simulations where the final populations again represent the average over 20 simulations with distinct random initial conditions. Increasing the mutation probability progressively (note the logarithmic horizontal axis) increases TFT’s final relative population, with the lattice being completely invaded at $\mu ≳10^{-2}$, until a breakdown of cooperative behavior occurs in the vicinity of μ≳0.8, which is reminiscent of the error threshold that replicating systems can sustain as was shown by Eigen [68]. The transition towards domination by TFT depends on both the mutation rate μ and the shape parameter *s*, while mutation rates exceeding a critical threshold (μ>0.8) further prevent generalized cooperation from emerging for a wide range of error rates.

Transition in the system behavior towards an invasion by TFT is however not only dependent on mutation probability but also on the magnitude of the correction $\Delta_{p}$ being applied. An increase in both parameters leads to an overall increase in the effect of mutations. Statistical ensembles were therefore analyzed while varying both parameters, for several different modes of the initial error rate ($\hat{p}\in \left\{10^{-1},10^{-2},10^{-3},10^{-4}\right\}$), again averaged over 20 simulations for each parameter value. A similar behavior was seen regardless of $\hat{p}$: when a threshold that depends on both *s* and μ is reached, a transition occurs after which TFT’s relative population fraction $f_{\mathrm{TFT}}\to 1$ (Figure 5B). As $\log_{10}\left[\mu \right]\to 0$ (i.e., μ→1, inset of Figure 5A) there is a significant decrease in cooperative behavior. There is therefore an *optimum* region in μ-*s* space that reliably leads to the emergence of cooperative behavior: further increase of the mutation rate past a critical threshold instead leads to this marked decrease in population fitness of cooperators. Remarkably, TFT nonetheless maintains complete dominance all the way up to this error threshold.

The final lattice state for five simulations taken at regular logarithmic intervals through the transition is shown in Figure 5C. Mean error distributions are shown in Figure 5D, where the dotted curve indicates the initial distribution of error rates for all species at t=0, and grey bars refer to the final distribution averaged over the last 25 iterations of the model. Colored curves denote the distribution specifics for ALLC, ALLD, and TFT. The transition towards an invasion of the lattice by TFT is concurrent with a marked decrease of its mean error rate as μ increases, while that of the other species remains mostly unchanged. Finally, Figure 5E shows population (colored curves) and stationarity (black curve) evolution—the fraction of lattice sites that changed strategies at each iteration (Equation (Equation 5) in the Appendix B)—for all species, while Figure 5F shows the mean error rate. Near the transition ($\mu ∼10^{-2}$), large fluctuations both in population dynamics and mean error rates can be seen and are reminiscent of behavior at the critical point of phase transitions [102].

## 5. Discussion

Origin of life theories must be able to explain, from a Darwinian perspective, the emergence of cooperation between unrelated agents to achieve an exhaustive description of the transition from non-living matter to biological complexity. We have shown that on a spatially extended system subject to stochastic perturbations, cooperation can definitely be favored against defectors when selection is coupled with heritability and variability. Moreover, this sudden invasion of the system in which the reciprocal strategy TFT dominates the system notwithstanding stochastic perturbations, which we term *cooperative takeover*, represents a prime example of emergent collective phenomena. Those results are consistent with the view put forth by Damer and Deamer that the emergence of life presupposes novel emergent functions relying on “network effects” [22]. Cooperative takeover also shares many characteristics of traditional phase transition phenomena such as a dramatic spatial reorganization on a scale comparable to the size of the system and the presence of large fluctuations in the neighborhood of the critical point (Figure 5). As has been frequently suggested, major evolutionary transitions, and in particular (the emergence of) life also presents many characteristics of critical phenomena [103,104,105,106,107]. Life thus appears as a spontaneous symmetry breaking, with the order parameter of the phase transition being suggested to be alternately organic molecules chirality [10], replication fidelity [16], or information exchanged [106].

Many important features of our results are directly dependent on the spatial medium onto which the IPD games are played. Formations of ALLC strategies being able to resist invasion by predatory ALLD are prime examples. In fact, even in a dynamic scenario the spatial grouping of strategies into multiple colonies, propagating through traveling wavefronts, allows cooperators to evade complete eradication by defectors against which they have no other defense mechanism (see Figure 6 for an example of such traveling waves in our simulations). Correspondingly, diverse models of hypercycles and RNA cooperators have been shown to survive more easily in two-dimensional spatial settings compared to well-mixed systems [43]. In our model, randomizing the neighborhood of the strategies—thus evolving the systems precisely in “well-mixed” environments—prevents this takeover of TFT altogether.

A recurrent characteristic in many of these prebiotic scenarios is the importance of *thresholds* delimiting transitions between different levels of organizations. Their presence is well-known in biology—a prime example being Eigen’s error threshold discussed above. In the context of the origin of life theory, the transition towards the living state has indeed often been interpreted in terms of critical phenomena [10,16,106,107], while the concept of catalytic closure in autocatalytic sets [11,42] is rooted in that of critical transitions observed in random graph theory [108]. Specifically, in the prebiotic context, many other examples of thresholds have furthermore been suggested [96] such as that of spontaneous polymerization [109] or self-assembly of compartments [110]. In addition to the behavior expected at the error threshold for high values of the error rate μ included in our model, the results we have presented here clearly imply the presence of such a transition driven by an increase of the mutation rate past a critical value (c.f. Figure 5A), which ultimately leads to a predominance of the cooperation behavior assumed to be an essential characteristic of various scenarios for the transition towards the living state. While a precise characterization of this transition—namely to determine if it represents a true phase transition—remains to be established, it shares many of the common features associated with critical phenomena typically seen in physics.

Relatedly, information theory has often been invoked in explaining complexity [111,112,113]. Specifically, it has been hypothesized that major evolutionary transitions involve changes in the way information is stored and transmitted [103], and communication of information is a prerequisite for the emergence of cooperation [54]. The emergence of TFT as one of the simplest conditional strategies to *encode* also contributes to laying out the foundations for theories of life that, in the spirit of determining which features are universal in every possible living form, consider that its fundamental character lies in its capacity to process informational content [114,115,116]. If life is defined as “information that copies itself” then the origin of life problem becomes one of the origins of this information. Likewise, the problem that ought to be solved concerning the transition from the abiotic to fully-functional replicators encoding their own replication mechanism, in the context of models of the emergence of life on Earth such as the RNA world hypothesis, is an informational one. It is precisely this gap in the evolutionary history of the information content of early replicators that a minimal information-processing strategy such as TFT—whatever its precise physico-chemical implementation in a given early lifeform— could bridge.

## 6. Conclusions

Analyzing the origin of life problem, recent work has emphasized determining what elements could be considered universal in our definition of life, as opposed to contingent ones such as the specific substrate or precise evolutionary history of life on Earth [10]. The presence of abiotic cooperation that preceded and subsequently allowed more complex biological organisms to evolve is most likely one of those universal elements.

Using agent-based modeling of the iterated prisoner’s dilemma, we investigated the emergence of cooperative behavior in a stochastic and spatially extended setting and characterized the effects of inheritance and variability. Our results demonstrate that there is a mutation threshold above which a dramatic and robust cooperative takeover of the system takes place in a manner reminiscent of a phase transition, and that the simplest conditional strategy “Tit-for-Tat” plays a key role in the process.

Generally speaking, one would tend to expect that the emergence and sustenance of cooperation in an evolutionary context would require a reasonable level of stability regarding the environment in which interaction between cooperating entities should take place. Stability here may refer, among other factors, both to a sufficient level of fidelity as to the mutual interactions between replicators, whatever the exact nature of these early interactions may be, and to a reliable inheritability of the features favoring cooperative behavior, whatever the mechanisms of early replication involved.

Our simulations results paint an altogether different picture with regard to the impact of stochastic perturbations. Despite the presence of high error rates in interaction and the contribution of stochastic processes in the dynamics of heritability, the emergence and predominance of cooperators are sustained all the way up to the Eigen error catastrophe. Placed in the context of current origin-of-life ideas and taken at face value under the assumption that cooperation is an essential building block of complexity, our simulations generally support the idea that the emergence of life—or at the very least of biochemical complexity—is a robust phenomenon that is not so easily disrupted by external perturbations, a result in agreement with the suggestion by Damer and Deamer that progenotes would have been selected for their stability [22]. Specifically, environments incurring high fluxes of ionizing radiation from very active host stars or exoplanetary surfaces subjected to strong environmental perturbations from tectonic or volcanic activity may not pose such strong constraints on the emergence of life—which is likewise qualitatively consistent with the early appearance of life on Earth.

## Figures and Tables

**Figure 1 life-12-00254-f001:**
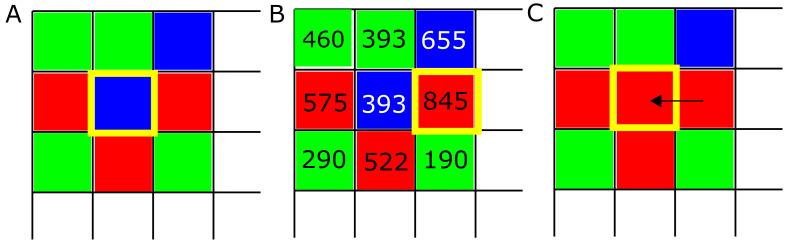
Evolution algorithm. (**A**) At the beginning of each simulation, strategies ALLC (“always cooperate”, green), ALLD (“always defect”, red), and TFT (“mutual reciprocation”, blue) are placed randomly on the lattice. Then the following process is repeated at each lattice site: strategies play *M* PD games against every neighbor in their neighborhood, and the scores of each player are recorded. (**B**) In a second step, the score of each strategy is compared against their neighbor’s score, and (**C**) the highest scoring strategy (boxed in yellow) is assigned to the site being examined. This process is repeated for each of the *T* iterations of the model.

**Figure 2 life-12-00254-f002:**
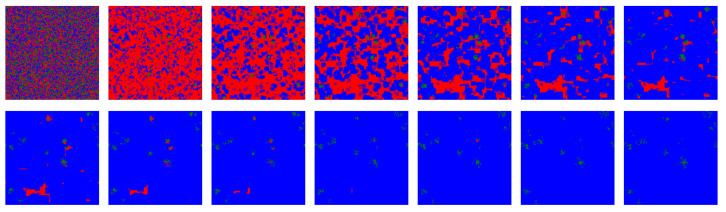
First fourteen iterations of the model for a deterministic simulation (i.e., p=0): strategies ALLC (green), ALLD (red), TFT (blue), and RND (“play randomly”, pink) are initially placed randomly on the lattice, then IPD games are played and the highest scoring strategies are propagated. For such a simulation where no stochasticity is included—each move is determined by the assigned strategy and the players make no mistake playing the PD—a very brief invasion by ALLD precedes an eventual predominance of TFT. RND is quickly eliminated from the lattice, as in most of the simulations over the parameter space of the model. The majority of such deterministic simulations become stationary after only a few tens iterations.

**Figure 3 life-12-00254-f003:**
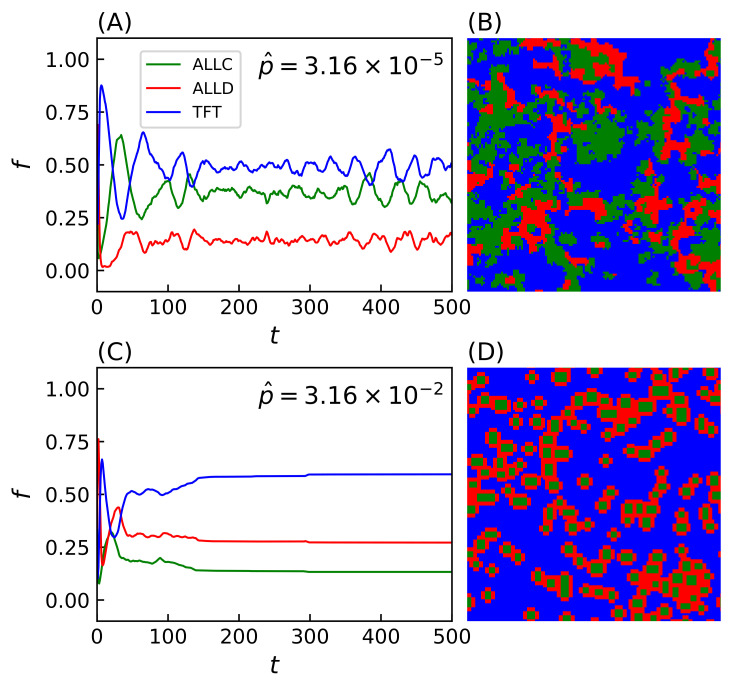
Population evolution and final lattice state for two simulations with $\hat{p}≃3.16\times 10^{-5}$ (**A**,**B**) and $\hat{p}≃3.16\times 10^{-2}$ (**C**,**D**). Simulations were carried out on a Cartesian lattice of size L=128 with periodic boundaries over T=500 iterations of the model, with iterated prisoner’s dilemma (IPD) games of M=2000 moves. In the second lattice (**D**), formations of ALLC cooperating together successfully survive while being surrounded by defectors. An animation of the two simulations is available in the Appendix A of the online version of this article.

**Figure 4 life-12-00254-f004:**
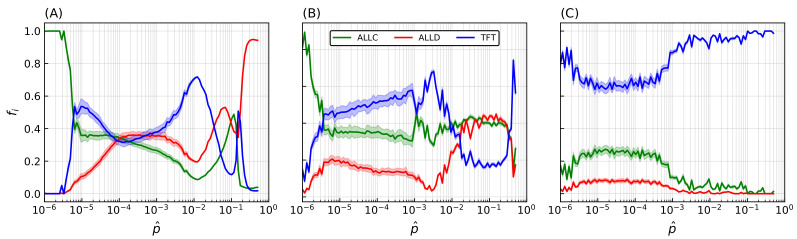
Final populations fractions for error rates that are immutable, heritable, and subject to mutations. Relative population frequencies with respect to the total population of strategies following either a strategy of unconditional cooperation (ALLC), unconditional defection (ALLD), or mutual reciprocity (TFT), as a function of the most probable error rate of the initial distribution $\hat{p}$. Final population fractions are averaged over samples of 20 simulations with random initial conditions, with shaded areas proportional to the standard deviation of final populations. Simulations were all carried out on a 2D periodic lattice of side length L=128 sites over T=500 iterations of the model, with IPD games of M=2000 moves. Error rates were initially set according to a lognormal distribution with the most probable values in abscissa and a fixed shape parameter s=0.5. (**A**) Error rates are immutable over the duration of the simulation. (**B**) Error rates become heritable (i.e., the player losing the game adopts its opponent’s error rate in addition to its strategy). (**C**) Introduction of a mutation probability $\mu =10^{-4}$. While heritability readily contributes to an increase in cooperative behavior—mostly at lower (≲$10^{-3}$) error rates—the introduction of mutations (**C**) is the primary driver of the dominance of cooperative behavior, allowing TFT to invade the lattice on a much larger region of the system’s parameter space.

**Figure 5 life-12-00254-f005:**
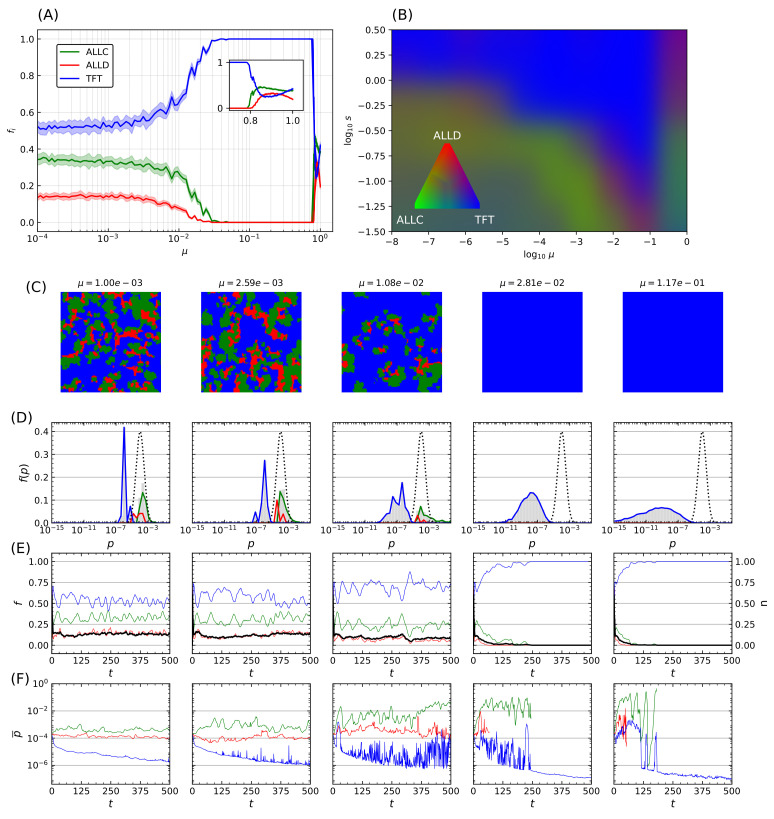
Transition towards TFT-mediated cooperation. (**A**) Relative population frequency for each strategy as a function of the mutation rate μ. Simulation parameters are identical as in Figure 4, and the data presented here refers to simulations initialized with an initial distribution of error rates having $\hat{p}=10^{-4}$ and s=0.5. Final populations are averaged on ensembles of 20 simulations with random initial distributions of strategies, with shaded areas proportional to the standard deviation in final population fractions. Increasing the mutation probability μ drives a continuous transition of the final population fractions towards invasion by TFT until a breakdown of cooperative behavior occurs at very high error rates (inset). (**B**) RGB-coding for relative population frequency of mutual cooperative behavior (TFT) as a function of mutation rate μ and shape parameter *s* of the error rate distribution. Measures are derived from 100 samples of *s* and μ combinations, and the results for each parameter configuration pair are averaged over 10 simulations with random initial conditions. (**C**–**F**) Final lattice state, error rates distributions, and temporal population, stationarity index, and mean error rate evolutions of five representative simulations taken from the statistical ensemble shown on panel (**A**). The initial distribution of error rates is shown as a dotted curve on (**D**), grey bars indicate the error rates distribution for all strategies, while colored curves indicate the ones specific for each strategy. Panel (**E**) shows population evolution with colored curves, while the black curve indicates the stationarity of the simulation. Panel (**F**) displays the mean of the error rate for each strategy.

**Figure 6 life-12-00254-f006:**
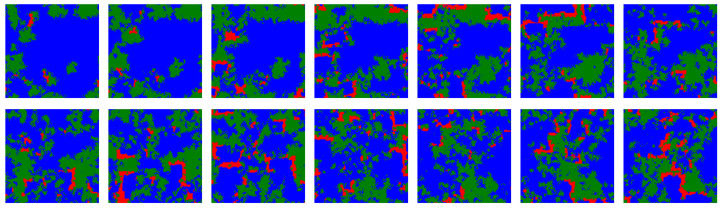
Temporal snapshots from a simulation including ALLC (green), ALLD (red) and TFT (blue) with initial error rate distribution parameters $\hat{p}=10^{-6}$ and s=1, and for which $\mu =10^{-4}$, taken at regular intervals of 25 iterations each. In this regime ALLC and TFT coexist while ALLD is a predator for ALLC. Traveling wavefronts of ALLC propagate through time on the lattice, although the fragmentation of ALLC populations into spatially distinct colonies acts as a barrier guarding against complete and immediate invasion by ALLD. The dynamics of the three strategies is reminiscent of a forest-fire model where there is an accumulation of trees (ALLC) according to a growth rate dictated by simulation parameters, which is conducive to dramatic reorganizations of the lattice following ignition (ALLD mutants).

**Table 1 life-12-00254-t001:** Score matrix of the Prisoner’s dilemma (PD). Numbers refer to the PD’s score matrix as traditionally defined [57], indicating the reward of the player adopting the strategy in the leftmost column. Mutual cooperation leads to the best possible mean outcome, while defection either leads to the absolute maximum or the absolute minimum reward. Letters refer to the general form of the score matrix for the PD: mutual cooperation leads to the reward payoff (R), while mutual defection leads to the punishment (P). Exploitation—with one player defecting while the opponent chooses to cooperate—leads to the temptation payoff (T) and sucker’s payoff (S). The game satisfies the PD constraint when T>R>P>S.

	Cooperate		Defect	
Cooperate		R		S
	3		0	
Defect		T		P
	5		1	

## Data Availability

Not applicable.

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
