# Peer review of "A Mutation Threshold for Cooperative Takeover"

_life, 2022, doi:10.3390/life12020254_

Round 1

Reviewer 1 Report

Champagne-Ruel and Charbonneau examined a specific evolutionary game, iterated prisoner’s dilemma (IPD), by introducing an “error rate” (which is a probability of choosing an unassigned strategy), a “heritability” (of the error rate), and a “mutation” (that changes the error rate). IPD games were performed based on a 2D-lattice model, and the authors found that a particular evolutionary strategy, Tit-for-Tat (TFT), tended to dominate the population as a mutation rate increased. At relatively high mutation rates, TFT became always selected, and the transition toward TFT was called “Cooperative Takeover” based on the nature of TFT (a player replicates an opponent’s action). I believe that the results are interesting as research of evolutionary game theory and possibly give insights into the origin and early evolution of life. A unique point is that the study tried to apply a game-theoretical framework of evolutionary biology into origins of life. However, in the current manuscript, the relationships between the simulation results and the origins of life are very ambiguous and how their results could contribute to the understanding of the origins of life should be clarified. I suggest a major revision based on the following comments.

  1. I did not clearly understand the relationships between the model with the origins of life, which should be more clarified in the Introduction section and elsewhere. What kind of replicators did the authors assume (e.g., a particular ribozyme)? What kind of conditions does the 2D surface represent (e.g., mineral surfaces)? What kind of situations do IPD and the iterated interactions with only nearby neighborhoods resemble (I do not come up with an example)? Is TFT a realistic strategy for a primitive replicator? I do not intend to argue that these assumptions are wrong, but I recommend the authors provide clearer explanations that connect the model to a likely (or at least possible) situation of the origins of life.

     I understand that the authors tried to explain some of the above concerns in the manuscript. For example, the authors argued at L94 that “the PD studied in game theory also applies to contexts where the players possesses no capacity for rational reasoning, and is a useful framework in analyzing biochemical cooperation phenomena in the context of prebiotic chemistry.” However, this argument is weak because it is based solely on biological organisms and lacks concrete examples related to the origins of life. The authors also wrote at L291 that “While such lengthy interactions…, they are nonetheless compatible with the hypothesis put forth by Damer that “Warm Little Ponds” subject to wet-dry cycling could have promoted rapid polymerization and longer polymer chains”, but I do not find a specific relevance between interactions among replicators and condensation of nucleotides for polymerization. At L298, the authors questioned “Is a biochemical counterpart for the PD strategies included in the simulations presented here really plausible?”, but again the answer is based on the Axelrod’ tournament and sophisticated biological organisms. If the authors could come up with a specific prebiotic situation, to which the model can be applied, that would make the study much more insightful to the origins-of-life field. The authors should also emphasize the relevance between the study and the origins of life in the Introduction section because, without that, readers can only understand the importance of the study when reaching the Discussion section. Finally, once reading up to L387, the authors said that “We realize and fully acknowledge that the exact relationship between our lattice IPD simulations and any specific physical/prebiotic or early biotic system is difficult to establish with confidence.” I believe this statement significantly weakens the study’s contributions to the understanding of the origins of life, but if the authors intend to “seeks to identify universal patterns and mechanisms that are independent of specific physical/chemical contexts and substrates”, please describe the motivation in Introduction.

     The following paper may help clarify how the study could contribute to the understanding of the origins of life because it employed a game-theoretical approach to examine interactions between reproducing ribozymes in the context of the origins of life. A score matrix of PD was also observed in an experiment.

     Yeates, J. A., Hilbe, C., Zwick, M., Nowak, M. A., & Lehman, N. (2016). Dynamics of prebiotic RNA reproduction illuminated by chemical game theory. Proceedings of the National Academy of Sciences, 113(18), 5030-5035.

  1. Several paragraphs are seemingly irrelevant to Conclusions. For example, the paragraph “In the prebiotic context ~” discussed the influence of thermodynamic fluctuations on prebiotic chemistry. The authors may have meant the importance of environmental perturbations as a driver for high error rates, but the statement was unclear. Similarly, the paragraphs “This robustness…” and “Far from being…” also seem irrelevant to the research, and they are clearly not conclusions.

  1. (Fig. 4A and B) A very high error rate (e.g., p ~ 0.5) means almost complete randomness. In this case, each strategy becomes seemingly identical, but a particular strategy (ALLD in Fig. 4A and TFT in Fig. 4B) dominated the population. On the other hand, a very low error rate essentially means no error, but the result with the minimum p value (the dominance of ALLC) contrasted with that with p=0 (Fig. 2, the dominance of TFT). Could the authors explain the underlying mechanisms for these observations? Are they because of variations in error rates depending on “s”, and the minimum p is non-negligible? Some discussions should help readers interpret the results correctly.

  1. About cooperative takeover, the authors claimed it based on the dominance by TFT. Does TFT always cooperate after domination? When an error is permitted, TFT could be defective even after all TFT replicators employ cooperative strategies. Defection against a cooperative strategy is beneficial (T) and thus I think the strategy could spread at least tentatively. What is a ratio of cooperative and defective strategies of TFT after it dominated the population?

  1. (L209) The authors claimed that “While system behavior at ^p>=10^(-3) does not differ significantly from previous simulations”, but looking at Fig. 4, the behavior of Fig. 4C significantly differs from that of Fig. 4A or B. Relatedly, the dominance of TFT at a high error rate could be explained by “Error rates are thus driven towards lower values when mutations are allowed, which ultimately leads to the complete invasion of the lattice by TFT”, but why did TFT also dominate the population at low error rate in contrast to Figs. A and B?

  1. 3 and Fig. 4 are probably in opposite positions.

  1. (Fig. 6) Please specify the mapping between colors and strategies

Reviewer 2 Report

Dear Editor, dear Authors,

the manuscript 

'A Mutation Threshold for Cooperative Takeover'

represents an interesting manuscript that is relevant for the Origin(s) of Life / Prebiotic Chemistry community. I recommend publication after revising it according to my small comment below.

COMMENTS

Figure 4
Refine label overlaps on the x-axis

Round 2

Reviewer 1 Report

I see clear improvements in the revised manuscript. The authors have clarified where the research stands in the field of the origins of life. I do not have further major comments. A very minor thing: please look over the list of references because they miss key information such as years of publication (which apparently happened during revision).
